# Clever-guided Robustness-aware Head Pruning For Transformer Models

## Abstract

Transformers lie at the core of modern AI, yet their exposure to adversarial perturbations raises reliability concerns. Empirical defenses often lack guarantees, while certification-based approaches provide them at nontrivial computational cost. We introduce RAHP (Robustness-Aware Head Pruning), a CLEVER-guided pruning framework for Transformers. RAHP scores each attention head with a composite of (i) Fisher information, the estimated accuracy cost of removing it. We prune heads that maximize robustness gain per accuracy cost, and (ii) $\Delta$CLEVER, an estimated increase in a CLEVER-style robustness lower bound when masking that head. Across evaluated tasks, RAHP yields compact models with higher CLEVER-style robustness scores, minimal change in clean accuracy, and improved resistance to both human-crafted and automatic adversarial attacks, all without adversarial retraining. Importantly, we use CLEVER as an *estimated* robustness proxy rather than a formal certificate, and our claims focus on improving CLEVER-style robustness metrics rather than providing new certified guarantees.

## 1 Introduction

The field of artificial intelligence has witnessed a paradigm shift with the emergence of transformer architectures, which have revolutionized not only natural language processing but also computer vision, speech recognition, and numerous other domains. However, as these models become increasingly common in critical applications, a fundamental question emerges: **how robust are these systems when faced with unexpected inputs, or adversarial attacks?**

In recent years, research has revealed critical vulnerabilities in Transformer-based models, demonstrated through concrete adversarial attacks that exploit these weaknesses. These attacks are often invisible to human readers: small changes at the token or character level, harmless word substitutions, or reworded sentences may appear trivial, yet they can cause a model to make drastically different decisions. These manipulations occur at various levels. At the character level, minor edits (e.g., "hotel" → "h0tel") preserve readability but can mislead the model (Ebrahimi et al., 2017). At the word level, replacing a word with a semantically similar alternative (e.g., "terrible" → "awful") retains the meaning for humans but alters the model's internal representation (Gan et al., 2021). At the sentence level, paraphrasing (e.g., "The movie was surprisingly good." → "I was taken aback by how enjoyable the film was.") maintains the intended message but may shift the model's response (Krishna et al., 2023). Such perturbations can lead to changes in embedding vectors, noisy or misleading representations, out-of-vocabulary tokens, or fragmented tokenization, which ultimately result in unstable or incorrect model behavior.

*Robustness* refers to a model's ability to maintain reliable performance in the face of variations, imperfections, or challenges in the input data (Freiesleben & Grote, 2023). This includes resilience to noise, shifts in data distribution, and deliberate adversarial manipulations (Brown et al., 2023). Broadly speaking, robustness can be categorized into three key types. *Adversarial robustness* concerns the model's ability to resist carefully crafted perturbations designed to cause incorrect outputs, even when the changes are imperceptible to humans (Shao et al., 2021). *Distributional robustness* focuses on how well a model generalizes when the test data distribution deviates from the training distribution, a common challenge in real-world deployment (Samuel & Chechik, 2021). *Certified robustness*, in contrast, offers formal guarantees. It quantifies the maximum perturbation under which a model's prediction is guaranteed to remain unchanged, typically using techniques

rooted in provable bounds or integer programming methods (Zeng et al., 2023; Kumar et al., 2023). While adversarial and distributional robustness are essential in practice, they often rely on empirical testing or assumptions about the nature of future inputs. Certified robustness provides a rigorous, model-agnostic foundation for developing robustness-oriented methods, making it a principled and compelling focus for algorithmic design and research.

Building on these three categories, the mechanisms to achieve robustness differ in emphasis. For adversarial robustness, the dominant recipe is to expose the model to worst-case perturbations during training (adversarial training) and/or add sensitivity-controlling penalties such as input-gradient regularization or virtual adversarial training that smooths the output locally around each sample (Madry et al., 2017; Ross & Doshi-Velez, 2018). For distributional robustness, methods optimize for performance under plausible distribution shifts rather than single samples, e.g., group DRO to raise worst-group accuracy and invariance-seeking objectives such as Invariant Risk Minimization (IRM) for OOD generalization (Sagawa et al., 2019; Arjovsky et al., 2019). For certified robustness, training and evaluation rely on provable bounds that guarantee prediction invariance within a perturbation set, via randomized smoothing, or convex relaxations of the adversarial region (Cohen et al., 2019; Wong & Kolter, 2018); metrics such as CLEVER (Weng et al., 2018) provide attack-agnostic estimated lower bounds that are widely used to assess robustness even when certification is not directly optimized. A key strength of certified approaches is that their guarantees are independent of any particular attack strategy, enabling model and threat-agnostic comparisons across methods.

In this work, we rely on CLEVER only as an *estimated* robustness lower bound, not as a formal certificate. Its value depends on Monte Carlo sampling and local Lipschitz approximations, so the guarantees it suggests are approximate and can be loose or optimistic in some regions of the input space. Thus, throughout the paper, we interpret CLEVER as a robustness *proxy*, and our claims focus on improving CLEVER-style robustness scores, rather than providing new provable guarantees.

*Pruning* removes parameters, neurons, or components (e.g., attention heads) to simplify the network while preserving accuracy. Beyond efficiency, pruning can also shape decision geometry by reducing unstable non-linearities and eliminating fragile pathways, which has been observed to improve empirical robustness, especially when combined with robust training objectives (Sehwag et al., 2020). Recent work further shows that pruning can improve certified robustness: relaxing or removing unstable ReLUs can tighten verification bounds and raise certified accuracy, and strategically grafting linearity in place of weak non-linear units likewise boosts certifiable guarantees (Zhangheng et al., 2022; Chen et al., 2022). In this spirit, we extend robustness-aware pruning to Transformers: we use the CLEVER score (Weng et al., 2018) as a guiding, certification-oriented signal alongside accuracy-preservation criteria to rank attention heads for removal, and show across tasks that targeting pruning by a CLEVER-style robustness metric increases a model's estimated robustness while minimally affecting clean performance.

In this work, we translate this certification-inspired, robustness-guided pruning idea into practice by introducing RAHP: Robustness-Aware Head Pruning. RAHP treats every attention head as a candidate for removal and scores it using two complementary signals: (i) Fisher information, which estimates the accuracy loss incurred if the head is pruned, and (ii) $\Delta$CLEVER, which estimates the change in the CLEVER-style robustness proxy when masking that head. By pruning the heads that maximize a weighted trade-off between these scores, RAHP compresses the model while increasing its CLEVER-style estimated perturbation radius. Because the CLEVER score is embedded directly in the pruning rule, RAHP directly integrates a certification-inspired robustness proxy into a structural pruning objective, jointly targeting robustness and efficiency **without** costly adversarial retraining and with negligible impact on clean accuracy. We do not claim new formal certificates; instead, we use CLEVER to steer pruning toward configurations that empirically improve robustness under both CLEVER-style metrics and adversarial attacks.

Concretely, this work makes the following contributions: (i) we propose RAHP, a head-pruning framework that combines Fisher-based accuracy costs with CLEVER-based robustness rewards to select attention heads; (ii) we conduct an extensive empirical evaluation on multiple Transformer backbones, GLUE and AdvGLUE benchmarks, and additional black-box attacks, showing that RAHP improves robustness while preserving clean accuracy under substantial pruning; and (iii) we provide a detailed analysis of the resulting pruning patterns and an ablation study of the trade-off parameters, illustrating how certification-inspired robustness proxies can effectively guide structural pruning.

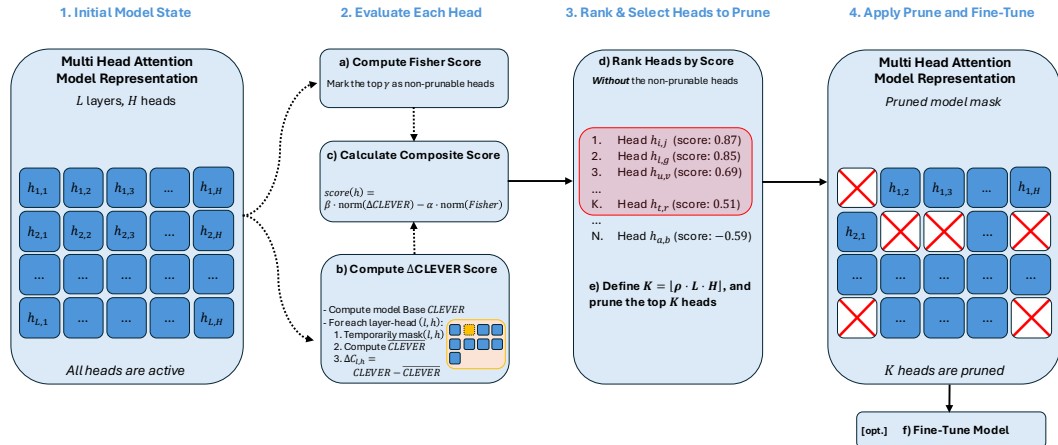

Figure 1: **RAHP Method Overview:** For each head, (a) compute Fisher score, (b) compute ΔCLEVER score, and (c) combine into a composite score. (d) Mark the top-$\gamma$ heads by Fisher as non-prunable and rank the remaining heads by composite score. (e) Prune the top-$K$ heads.

## 2 RELATED WORKS

Prior work strengthens Transformer robustness by smoothing predictions or explicitly optimizing against perturbations. R-Drop enforces output consistency across different dropout masks via a bidirectional KL term, reducing variance and sharpening decision margins (Wu et al., 2021). Child-Tuning masks gradients to update only a "child" subset of parameters, stabilizing fine-tuning on limited data and improving robustness without overfitting the full model (Xu et al., 2021). SMART augments fine-tuning with small, principled adversarial perturbations plus a KL smoothness regularizer in representation space (Jiang et al., 2019). FreeLB performs multi-step adversarial training in the embedding space to craft stronger perturbations while accumulating informative gradients (Zhu et al., 2019). While these approaches improve robustness during fine-tuning, they do not provide attack-agnostic guarantees, nor do they exploit structural properties of Transformer architectures. This contrasts with our approach, which leverages a certification-inspired robustness proxy to guide structural pruning decisions.

Another line of work pursues certified robustness with formal guarantees. Randomized smoothing wraps a base classifier with Gaussian noise to certify an $\ell_2$ radius per input (Cohen et al., 2019), while symbolic/interval bound propagation adapts certification to NLP by bounding the worst-case effect of discrete edits such as synonym or character substitutions (Huang et al., 2019). CLEVER, in turn, estimates attack-agnostic lower bounds on the minimal adversarial distortion and is widely used as a certification-oriented assessment metric even when certificates are not explicitly computed (Weng et al., 2018). However, CLEVER remains an estimated lower bound and does not constitute a formal certificate; its role in this work is therefore as a scalable robustness proxy rather than a definitive guarantee. These approaches clarify the trade-offs between guarantees, accuracy, and computational cost. More advanced formal certification methods (Brix et al., 2024; Bonaert et al., 2021) go beyond CLEVER and provide stronger, more precise guarantees, but they are typically orders of magnitude more expensive and have not yet been widely adopted for large NLP models. Our approach is therefore complementary: rather than competing with state-of-the-art verifiers, we use CLEVER as a lightweight, certification-inspired signal for ranking attention heads during pruning.

Pruning has emerged as a structural route to robustness by removing fragile pathways or reducing unstable non-linearities. Beyond empirical gains, recent studies show pruning can improve certified robustness by tightening verification bounds and reducing neuron instability (Zhangheng et al., 2022; Chen et al., 2022). In parallel, ROSE (Robust Selective Fine-Tuning) selectively updates robust parameters during adaptation, improving adversarial robustness and offering a strong baseline for comparison in NLP (Jiang et al., 2022).

# 3 METHODOLOGY

As illustrated in Figure 1, we propose a two-score pruning framework that, in a single global step across all layers, removes self-attention heads to enhance robustness without sacrificing accuracy.

Concretely, each attention head is assigned two complementary scores: a **CLEVER**-based robustness score that reflects the change in robustness when the head is masked, and a **Fisher**-based importance score that quantifies the accuracy cost of its removal. After normalization, the two signals are combined into a composite score, and all heads are ranked globally. To safeguard accuracy, a fixed fraction $\gamma$ of the most Fisher-important heads are explicitly protected from pruning. The remaining heads are pruned according to the target pruning ratio $\rho$, removing those with the highest composite scores. Finally, the pruned model can be optionally fine-tuned slightly on the training data to mitigate potential clean performance loss.

## 3.1 NOTATION

We define the notation used throughout this paper as follows. Let $f(\cdot)$ denote a Transformer-based sequence classifier. The model produces logits $z = f(x) \in \mathbb{R}^C$ corresponding to $C$ output classes.

The Transformer model consists of $L$ attention layers, where each layer $\ell \in \{0, \ldots, L-1\}$ contains $H$ attention heads. A specific head is identified by the tuple $(\ell, h)$, with $h \in \{0, \ldots, H-1\}$. A binary head mask is defined as $M \in \{0,1\}^{L \times H}$, where $M_{\ell,h} = 0$ indicates that the head $(\ell, h)$ is pruned, and $M_{\ell,h} = 1$ indicates that it is retained. Finally, the classification margin for a given output is defined as $g(z) = z_c - z_{c'}$, where $z_c$ is the logit of the correct class, and $z_{c'}$ is the highest logit among all incorrect classes.

## 3.2 CLEVER-BASED ROBUSTNESS SCORE

To quantify model robustness, we adapt a CLEVER-style estimate(Cross-Lipschitz Extreme Value for nEtwork Robustness) score (Weng et al., 2018). The CLEVER score provides a lower bound on the minimum perturbation needed to cause a misclassification, with higher scores indicating greater robustness. It is formally defined as the minimum ratio of the classification margin to the norm of its gradient with respect to the input, approximated via extreme value theory.

We estimate this score for a batch of samples, where the score for a single input $x$ is:

$$C(x, M) = \frac{g(f(x, M))}{\|\nabla_x g(f(x, M))\|_2 + \epsilon}. \tag{1}$$

Here, $f(x, M)$ is the model's forward pass using the head mask $M$, and $\epsilon$ is a small constant for numerical stability. The gradient is taken with respect to the input embeddings. Note that Equation 1 is a *local CLEVER-style approximation* computed on sampled inputs; it does not constitute a formal certificate. Throughout, we interpret higher values of $C(x, M)$ as indicating stronger robustness *under this estimator*, rather than a provable guarantee.

To assess the contribution of an individual head $(\ell, h)$, we measure the change in CLEVER score when this head is removed. Formally, letting $M$ be the current mask and $M'_{\ell,h}$ the same mask but with head $(\ell, h)$ pruned, we define the **Robustness Score** as:

$$\Delta C_{\ell,h} = \mathbb{E}_{x \sim D} \left[ C(x, M'_{\ell,h}) - C(x, M) \right] \tag{2}$$

where the expectation is taken over the dataset $D$.

The interpretation of $\Delta C_{\ell,h}$ is straightforward:

- If $\Delta C_{\ell,h} = 0$, pruning head $(\ell, h)$ has no effect on robustness.
- If $\Delta C_{\ell,h} > 0$, the model becomes more robust after pruning $(\ell, h)$, making them good pruning candidates.
- If $\Delta C_{\ell,h} < 0$, pruning $(\ell, h)$ weakens robustness and such heads are best kept.

In practice, the higher the $\Delta C_{\ell,h}$ value, the more robust the pruning of head $(\ell, h)$ contributes to the overall model. This makes $\Delta$CLEVER a natural robustness-oriented reward signal within our composite pruning criterion. In practice, we approximate the expectation over $x \sim \mathcal{D}$ with an average over a randomly sampled subset of the data points, which keeps the scoring pass computationally manageable while still capturing the relative robustness contribution of each head.

### 3.3 FISHER-BASED ACCURACY SCORE

To prevent the pruning process from significantly degrading model accuracy, we must quantify the importance of each attention head to the model's primary task. For this, we use the *Fisher Information* (FI), which measures the sensitivity of the model's loss to changes in its parameters. Because FI provides a principled sensitivity estimate, it has been widely adopted as a pruning criterion (Molchanov et al., 2016; Theis et al., 2018; Molchanov et al., 2019; Liu et al., 2021; Kwon et al., 2022; Sung et al., 2021). Heads with high FI are considered more critical to the model's performance, as pruning them would likely incur a large accuracy cost.

We approximate the diagonal of the FI matrix for the parameters $\theta_{\ell,h}$ of each head $(\ell, h)$. For a single data point $(x, y)$, the FI is estimated as the squared gradient of the loss function $J$ (e.g., Cross-Entropy) with respect to the head's parameters:

$$F_{l,h}(x,y) = \left\| \nabla_{\theta_{l,h}} J(f(x), y) \right\|_2^2. \tag{3}$$

Averaging over the dataset $D$ yields the head-level Fisher estimate:

$$F_{l,h} = \mathbb{E}_{(x,y) \sim D} \left[ F_{l,h}(x,y) \right]. \tag{4}$$

For numerical stability and to temper the heavy-tailed distribution of Fisher values, we apply a log compression before scoring. The resulting $F_{\ell,h}$ acts as the **Accuracy Score**: heads with high Fisher values are strongly tied to minimizing task loss and should therefore be preserved, whereas heads with low Fisher values tend to contribute little to accuracy and are good candidates for pruning. As with the CLEVER-based score, we estimate $F_{\ell,h}$ on the same held-out subset of $\mathcal{D}$ used for robustness scoring, so that both accuracy and robustness signals are computed under comparable data conditions.

### 3.4 COMPOSITE SCORE & PRUNING RULE

After computing both the robustness and accuracy metrics for all heads, we combine them into a single signal that guides the pruning decision. Specifically, we first apply a log transformation to the Fisher estimates, $I_{\ell,h} = \log(F_{\ell,h} + \varepsilon)$, to stabilize their heavy-tailed distribution. We then normalize both $I_{\ell,h}$ and $\Delta C_{\ell,h}$ across all heads using min–max scaling to the range $[0, 1]$, obtaining the normalized accuracy cost $I'_{\ell,h}$ and robustness gain $\Delta C'_{\ell,h}$. This shared normalization ensures that the two terms are on comparable scales before forming the composite score.

The **Composite Score** $S_{\ell,h}$ for head $(\ell, h)$ is then defined as:

$$S_{\ell,h} = \beta \cdot \Delta C'_{\ell,h} - \alpha \cdot I'_{\ell,h}, \tag{5}$$

where $\alpha$ and $\beta$ control the trade-off between robustness and accuracy preservation. A higher value of $S_{\ell,h}$ indicates that pruning the head yields stronger robustness improvements (large $\Delta C'_{\ell,h}$) while incurring only a small accuracy penalty (low $I'_{\ell,h}$).

To ensure that accuracy-critical heads are not mistakenly removed, we protect a fixed fraction $\gamma$ of heads with the highest Fisher values (the *non-prunable set*). These are excluded from consideration regardless of their composite score. The remaining heads are globally ranked by $S_{\ell,h}$ in descending order, and pruning is applied to the top $K = \lfloor \rho \cdot L \cdot H \rfloor$ heads. Sorting in descending order ensures that we remove precisely those heads with the best trade-off: the highest robustness gain combined with the lowest accuracy cost. Put differently, the higher the composite score, the more it reflects a desirable balance of large robustness gains with minimal accuracy loss.

Empirically, we found that reserving a small non-prunable core stabilizes clean accuracy across tasks: heads with extremely high Fisher scores can occasionally exhibit modest robustness gains

---

**Algorithm 1** Robustness-Aware Head Pruning

---

**Input:** Model $f$; dataset $D$; weights $\alpha, \beta$; prune ratio $\rho$; non-prunable fraction $\gamma$
**Output:** Pruned head mask $M \in \{0,1\}^{L \times H}$

1: Initialize $M \leftarrow \mathbf{1}^{L \times H}$                *(all heads active)*
2: **for** $\ell = 0$ **to** $L - 1$ **do**
3:   **for** $h = 0$ **to** $H - 1$ **do**
4:    $M' \leftarrow M$ with $M'_{\ell,h} \leftarrow 0$
5:    $\Delta C_{\ell,h} \leftarrow \mathbb{E}_{x \sim D} \left[ \frac{g(f(x,M'))}{\|\nabla_x g(f(x,M'))\|_2 + \varepsilon} - \frac{g(f(x,M))}{\|\nabla_x g(f(x,M))\|_2 + \varepsilon} \right]$
6:    $F_{l,h} \leftarrow \mathbb{E}_{(x,y) \sim D} \left\| \nabla_{\theta_{l,h}} J(f(x), y) \right\|_2^2$
7:    $I_{\ell,h} \leftarrow \log(F_{\ell,h} + \varepsilon)$
8:   **end for**
9: **end for**
10: $I'_{\ell,h} \leftarrow \text{normalize}(I_{\ell,h})$           *(min-max scaling to [0, 1])*
11: $\Delta C'_{\ell,h} \leftarrow \text{normalize}(\Delta C_{\ell,h})$        *(min-max scaling to [0, 1])*
12: $\mathcal{N} \leftarrow \text{top-}\gamma$ fraction of heads with highest $I_{\ell,h}$    *(non-prunable heads)*
13: **for** $(\ell, h)$ over all heads **do**
14:   **if** $(\ell, h) \in \mathcal{N}$ **then**
15:    $S_{\ell,h} \leftarrow -\infty$           *(protect non-prunable heads)*
16:   **else**
17:    $S_{\ell,h} \leftarrow \beta \cdot \Delta C'_{\ell,h} - \alpha \cdot I'_{\ell,h}$
18:   **end if**
19: **end for**
20: $K \leftarrow \lfloor \rho \cdot L \cdot H \rfloor$
21: Prune top-$K$ heads with largest $S_{\ell,h}$ by setting $M_{\ell,h} \leftarrow 0$
22: Fine-tune $f$ on $D$ using the updated $M$        *(optionally)*
23: **return** $M$

---

under $\Delta C'_{\ell,h}$, but pruning them tends to disproportionally hurt task performance. Using a fixed $\gamma$ therefore provides a conservative safeguard for accuracy at negligible cost to robustness.

Finally, the pruned model can be optionally fine-tuned on the original dataset to recover any residual performance loss. This one-shot, global ranking strategy enables RAHP to jointly optimize robustness and accuracy without iterative per-layer pruning. The complete algorithm is summarized in Algorithm 1.

## 4 EXPERIMENTS

### 4.1 EXPERIMENTAL SETUP

**Datasets.** We evaluate our method on a broad set of text classification benchmarks, including four GLUE tasks: *SST-2*, *RTE*, *QNLI*, and *QQP* (Wang et al., 2018), and their adversarial counterparts from *AdvGLUE* (Wang et al., 2021). GLUE offers clean evaluation across sentiment analysis (SST-2), natural language inference (RTE, QNLI), and paraphrase detection (QQP), while AdvGLUE provides challenging adversarial variants that stress-test language models under high-quality adversarial perturbations.

To assess generalization beyond GLUE, we also include two additional large-scale datasets: *AG-News* for topic classification and *Yelp* for sentiment analysis.

**Attack Methods.** To complement the AdvGLUE evaluation, we also assess robustness under three popular black-box attacks from the `TextAttack` library (Morris et al., 2020). Their results are reported in Table 3. *BERT-Attack* (Li et al., 2020) generates fluent, contextually appropriate adversarial variants by leveraging a masked-language-model to propose high-quality substitutions. *TextFooler* (Jin et al., 2020) is a greedy synonym-substitution attack that iteratively replaces influential tokens until the model prediction flips. *TextBugger* (Li et al., 2018) combines synonym substitutions with lightweight character-level edits (e.g., swaps, deletions) to craft adversarial examples efficiently.

**Implementation Details.** Across all experiments, we fixed RAHP's pruning hyperparameters to values that provided the most stable and effective performance. Specifically, we set the trade-off weights to $\beta = 1$ and $\alpha = 0.5$, which we found to be the most effective combination for balancing

| Model | Pruning Volume | SST-2 | | RTE | | QNLI | | QQP | | Avg GLUE+AdvGLUE | Avg Δ↓ |
|---|---|---|---|---|---|---|---|---|---|---|---|
| | | GLUE | AdvGLUE | GLUE | AdvGLUE | GLUE | AdvGLUE | GLUE | AdvGLUE | | |
| **RoBERTa$_{BASE}$** | | | | | | | | | | | |
| Vanilla | 0% | 94.29 | 24.05 | 77.91 | 28.15 | 92.97 | 27.43 | 91.58 | 19.49 | 56.98 | 64.41 |
| R-Drop | 0% | 95.32 | 27.84 | 79.86 | 31.36 | 93.30 | 28.92 | 91.86 | 37.44 | 60.74 | 58.70 |
| CHILD-TUNING$_D$ | 70% | 94.21 | 23.82 | 75.52 | 16.54 | 92.36 | 31.89 | 91.64 | 17.95 | 55.49 | 65.88 |
| SMART | 0% | 94.98 | 35.95 | 77.54 | 24.44 | 93.35 | 34.29 | 91.04 | 46.58 | 62.27 | 53.91 |
| FreeLB | 0% | 94.89 | 35.81 | 78.42 | 32.10 | 93.12 | 36.22 | 92.04 | 44.10 | 63.34 | 52.56 |
| ROSE-First | 60% | 94.84 | 37.67 | 78.34 | 35.49 | 92.19 | 44.19 | 89.56 | 44.44 | 64.59 | 48.29 |
| ROSE-Second | 60% | 93.78 | 36.99 | 78.16 | 37.97 | 92.41 | 34.63 | 90.48 | 45.73 | 63.77 | 49.88 |
| ROSE-Ensemble | 60% | 94.09 | 39.36 | 78.63 | 38.02 | 92.64 | 39.59 | 90.39 | 47.44 | 65.02 | 47.84 |
| RAHP | 60% | 94.56 | 38.90 | 78.95 | 39.60 | 92.78 | 42.19 | 91.96 | 48.14 | **65.89** | **47.36** |
| **RoBERTa$_{LARGE}$** | | | | | | | | | | | |
| Vanilla | 0% | 96.08 | 56.08 | 85.92 | 61.73 | 94.58 | 63.38 | 92.09 | 40.60 | 73.81 | 36.72 |
| R-Drop | 0% | 96.59 | 53.38 | 85.56 | 66.67 | 95.01 | 55.95 | 92.35 | 44.80 | 73.79 | 37.18 |
| CHILD-TUNING$_D$ | 70% | 95.91 | 51.35 | 85.92 | 61.73 | 94.30 | 58.11 | 92.03 | 43.59 | 72.87 | 38.35 |
| SMART | 0% | 96.67 | 59.12 | 85.02 | 69.14 | 94.91 | 61.04 | 92.12 | 50.85 | 76.11 | 32.14 |
| FreeLB | 0% | 96.49 | 59.32 | 86.76 | 66.91 | 94.99 | 62.30 | 92.60 | 48.21 | 75.95 | 33.53 |
| ROSE-First | 60% | 95.58 | 57.77 | 85.13 | 70.62 | 94.08 | 64.02 | 90.67 | 60.26 | 77.27 | 28.20 |
| ROSE-Second | 60% | 96.29 | 60.59 | 85.08 | 67.49 | 94.72 | 63.68 | 91.68 | 55.90 | 76.93 | 30.03 |
| ROSE-Ensemble | 60% | 96.10 | 60.81 | 85.92 | 71.11 | 94.26 | 64.64 | 91.46 | 60.51 | 78.10 | 27.67 |
| RAHP | 60% | 96.32 | 62.02 | 85.96 | 71.25 | 94.74 | 67.51 | 90.31 | 57.12 | **78.15** | **27.36** |

Table 1: Accuracy on GLUE and AdvGLUE benchmarks. The last column shows the drop from GLUE to AdvGLUE (lower is better). Bold indicates the best result.

robustness and importance. In our pruning configuration, we remove 60% of the attention heads (leaving only 40% active) for the experiments reported in Tables 1 and 2. For Table 3, we used a 40% prune ratio. In addition, we enforced a non-prunable fraction of $\gamma = 10\%$ for all experiments. More details can be found in Section A.

## 4.2 EVALUATION ON GLUE AND ADVGLUE BENCHMARKS

We compare RAHP to all baselines on SST-2, RTE, QNLI, and QQP tasks from GLUE and AdvGLUE. All results are averaged over five random seeds. Table 1 summarizes the results:

(1) *RAHP substantially improves robustness on AdvGLUE.* Across all four tasks, RAHP achieves the best or near-best adversarial accuracy, outperforming vanilla fine-tuning and strong regularization-based baselines such as R-Drop and CHILD-TUNING$_D$, as well as adversarial-training methods like SMART and FreeLB.

(2) *RAHP maintains competitive clean accuracy under high pruning.* Despite operating at a high pruning volume, RAHP maintains GLUE accuracy comparable to other strong baselines, or only slightly below the best results, indicating that robustness gains do not come at the cost of significant degradation on clean inputs.

(3) *RAHP offers the best trade-off between clean and adversarial performance.* The average GLUE → AdvGLUE drop reported in the last column of Table 1 is smallest for RAHP, showing that our method most effectively reduces the gap between standard accuracy and adversarial robustness.

Table 2 extends the evaluation of RAHP to four additional transformer backbones. Across all models, RAHP consistently achieves the highest overall average on SST-2 and RTE, for both GLUE and AdvGLUE. Averaging the four AUA scores for each backbone (SST-2 GLUE, SST-2 AdvGLUE, RTE GLUE, RTE AdvGLUE), RAHP outperforms the second-best method by $+1.22\%$, and surpasses the mean performance of all baselines by $+27.16\%$. For ACC, RAHP improves over all competing methods by $+3.5\%$. This shows that RAHP's substantial gains in adversarial robustness do not come at the expense of clean accuracy.

## 4.3 EVALUATION UNDER BLACK-BOX ADVERSARIAL ATTACKS

We further evaluate RAHP under three standard black-box word-level attacks: BERT-Attack, TextBugger, and TextFooler, implemented via the `TextAttack` library. For each defense, we report three metrics. *Clean%* denotes the classification accuracy on the original, unperturbed test set, and indicates how much a defense affects performance on normal inputs. *AUA%* (accuracy under attack) measures the accuracy on the adversarially perturbed test set and is our primary robustness metric. Finally, *#Query* is the average number of model queries the attacker needs to successfully generate an adversarial example; higher values correspond to a more attack-resistant model.

| Model | Method | SST-2 | | | | | | RTE | | | | | |
|---|---|---|---|---|---|---|---|---|---|---|---|---|---|
| | | | GLUE | | | AdvGLUE | | | GLUE | | | AdvGLUE | |
| | | AUA | ACC | AVG | AUA | ACC | AVG | AUA | ACC | AVG | AUA | ACC | AVG |
| DeBERTaV3$_{BASE}$ | ROSE | 15.81 | 84.17 | 49.99 | 39.50 | 90.76 | 65.13 | 19.10 | 73.00 | 46.05 | 32.09 | 78.26 | 55.18 |
| | FreeLB | 13.76 | **94.95** | 54.36 | 52.70 | **94.95** | 73.83 | 8.12 | 71.50 | 39.81 | **65.43** | **81.58** | **73.51** |
| | PEAFT | 34.17 | 94.61 | 64.39 | 64.58 | 94.61 | 79.60 | 55.24 | **80.14** | 67.69 | 59.82 | 80.14 | 69.98 |
| | RAHP | **37.22** | **94.95** | **66.09** | **65.91** | 94.68 | **80.30** | **56.90** | 79.31 | **68.11** | 58.96 | 80.08 | 69.52 |
| DeBERTaV3$_{LARGE}$ | ROSE | 19.65 | 78.46 | 49.06 | 52.61 | **95.50** | 74.06 | 22.00 | 72.98 | 47.49 | 70.50 | 83.71 | 77.11 |
| | FreeLB | 21.90 | 94.15 | 58.03 | 47.97 | 94.15 | 71.06 | OOM | OOM | OOM | OOM | OOM | OOM |
| | PEAFT | 42.32 | 95.30 | 68.81 | 68.64 | 95.30 | 81.97 | **58.42** | 88.81 | **73.62** | 72.71 | 88.81 | 80.76 |
| | RAHP | **44.60** | **95.55** | **70.08** | **68.93** | 95.28 | **82.11** | 58.13 | **88.91** | 73.52 | **73.80** | **89.43** | **81.62** |
| ELECTRA$_{BASE}$ | ROSE | 26.79 | 88.00 | 57.40 | 42.72 | 90.37 | 66.55 | 31.28 | 61.17 | 46.23 | 31.85 | **75.74** | 53.80 |
| | FreeLB | 11.58 | 93.81 | 52.70 | 44.59 | 93.81 | 69.20 | 12.10 | 69.01 | 40.56 | 43.21 | 74.00 | 58.61 |
| | PEAFT | **36.76** | **94.27** | **65.52** | 66.02 | 94.27 | 80.15 | 57.36 | 73.29 | 65.33 | 62.27 | 73.29 | 67.78 |
| | RAHP | 34.99 | 94.00 | 64.50 | **66.45** | **94.60** | **80.53** | 57.36 | 73.30 | 65.33 | **63.02** | 75.11 | **69.07** |
| ELECTRA$_{LARGE}$ | ROSE | 15.47 | 84.43 | 49.95 | 59.64 | 93.20 | 76.42 | 24.64 | 78.35 | 51.50 | **77.82** | 85.71 | **81.77** |
| | FreeLB | 8.60 | 95.41 | 52.01 | **63.51** | 95.41 | **79.46** | 7.58 | 81.23 | 44.41 | 69.13 | 81.22 | 75.18 |
| | PEAFT | **50.91** | 95.15 | 73.03 | 61.35 | 95.15 | 78.25 | 62.23 | 88.09 | 75.16 | 70.37 | 88.09 | 79.23 |
| | RAHP | 50.80 | **95.50** | **73.15** | 63.20 | **95.45** | 79.33 | **63.10** | **88.60** | **75.85** | 71.22 | **88.66** | 79.94 |

Table 2: Adversarial robustness on GLUE (under TextFooler attack) and AdvGLUE for multiple backbones, reporting accuracy under attack (AUA), clean accuracy (ACC), and their average (AVG).

The results for BERT$_{BASE}$ on SST-2, AGNews, and Yelp are summarized in Table 3. Overall, RAHP consistently preserves clean performance while substantially improving robustness and increasing the cost of successful attacks. On all three datasets, RAHP attains clean accuracy that is nearly identical to the best-performing baseline (MC2F), differing by at most 0.1–0.2 points, which confirms that our pruning strategy does not compromise standard accuracy.

In terms of adversarial accuracy, RAHP either matches or surpasses the strongest existing defense across most attack–dataset combinations. For BERT-Attack, RAHP achieves the highest AUA% on all three datasets, slightly improving over MC2F while requiring a comparable or larger number of queries. Under TextBugger and TextFooler, RAHP dominates earlier baselines such as FreeLB, WLRE, and SD, and is competitive with, or better than, MC2F: on AGNews and Yelp it achieves similar or higher AUA% while also increasing the average query counts, indicating that successful adversarial examples become more expensive to find.

Taken together, these results show that RAHP not only improves robustness on GLUE and AdvGLUE, but also yields a strong black-box robustness profile: it maintains near-optimal clean accuracy, consistently raises AUA% compared to prior defenses, and makes black-box attacks significantly more query-intensive.

| Dataset | Method | Clean % | BERT-Attack | | TextBugger | | TextFooler | |
|---|---|---|---|---|---|---|---|---|
| | | | AUA% | #Query | AUA% | #Query | AUA% | #Query |
| SST-2 | Fine-tune | **92.71** | 3.83 | 106.4 | 6.10 | 90.5 | 28.70 | 46.0 |
| | FreeLB | 92.01 | 23.88 | 174.7 | 29.40 | 132.6 | 49.70 | 53.8 |
| | WLRE | 92.11 | 29.80 | 185.4 | 32.80 | 138.4 | 50.10 | 56.4 |
| | SD | 91.36 | 36.46 | 201.2 | 46.30 | 167.3 | 54.50 | 62.3 |
| | MC2F | **92.71** | 40.05 | **289.4** | 52.60 | 184.2 | **61.80** | 84.2 |
| | RAHP | 92.25 | **40.33** | 285.1 | **53.85** | 189.3 | 60.69 | **84.5** |
| AGNews | Fine-tune | 94.68 | 4.09 | 412.9 | 14.70 | 306.4 | 40.00 | 166.2 |
| | FreeLB | 94.99 | 19.90 | 581.8 | 33.20 | 396.0 | 52.90 | 201.1 |
| | WLRE | 94.05 | 28.60 | 657.1 | 32.60 | 468.1 | 53.40 | 208.0 |
| | SD | 93.81 | 38.60 | 744.1 | 49.30 | 488.1 | 60.10 | 219.4 |
| | MC2F | **95.13** | **45.30** | **892.5** | 53.80 | 561.4 | 64.30 | 299.0 |
| | RAHP | 95.01 | **45.30** | 888.6 | **54.11** | 573.12 | **66.12** | **307.0** |
| YELP | Fine-tune | 95.19 | 5.40 | 116.2 | 5.20 | 105.4 | 29.60 | 52.6 |
| | FreeLB | 95.11 | 28.24 | 184.3 | 28.30 | 143.5 | 50.00 | 68.4 |
| | WLRE | 94.86 | 31.46 | 208.2 | 31.50 | 155.0 | 50.00 | 69.1 |
| | SD | 93.45 | 39.61 | 320.7 | 47.80 | 187.6 | 55.10 | 100.2 |
| | MC2F | **95.26** | 48.50 | 586.4 | **54.00** | 214.3 | 63.20 | 112.4 |
| | RAHP | 95.20 | **49.12** | **599.1** | 52.97 | 202.2 | **63.31** | **114.0** |

Table 3: Adversarial robustness of **BERT$_{BASE}$**: accuracy on the clean test set (Clean%) and under attack (AUA%), plus average queries per successful attack (#Query); higher (in bold) is better.

## 5 PRUNING BEHAVIOR ANALYSIS

### 5.1 LAYERWISE PRUNING PATTERNS

Figure 2 illustrates the layer-wise distribution of pruned heads in RoBERTa$_{\text{BASE}}$ on the SST-2 task across different pruning ratios. At a low pruning ratio of $10\%$, pruning is almost exclusively concentrated in the later layers, particularly from layer 9 onward. Increasing the ratio to $20\%$ preserves this pattern while extending pruning upward to layer 8, suggesting that robustness-aware pruning first targets redundancy in the deepest layers before affecting earlier components.

This trend becomes more pronounced as the pruning ratio grows. For ratios of $40\%$ and above, the distribution follows a clear structural pattern: layers 9–12 consistently absorb the highest pruning rates, often exceeding $80\%$ of their heads. From layer 8 downward, the proportion of pruned heads gradually decreases, reaching a minimum around layers 5–6. Interestingly, pruning volumes then rise again in the earliest layers (layers 1–4), indicating that shallow layers also contain removable redundancy once the deeper layers have been heavily pruned.

Overall, these results reveal a non-uniform pruning distribution: robustness-aware pruning strongly favors removing heads in the deepest layers, followed by shallow layers at high pruning volumes, while the middle layers are relatively more protected. This layered trend aligns with prior observations that attention heads in later layers are more redundant, whereas middle layers encode features more central to task accuracy, as suggested in other papers (Ling et al., 2024; Zhang et al., 2024; Sajjad et al., 2023; Gromov et al., 2024).

### 5.2 INTERACTION BETWEEN ROBUSTNESS AND FISHER INFORMATION

To better understand RAHP's internal behavior, we visualize the per–head Fisher and CLEVER scores and examine how they interact in pruning. All results below are reported for RoBERTa$_{\text{BASE}}$.

Figure 3 shows side-by-side heatmaps of the normalized Fisher importance values and normalized $\Delta$CLEVER scores. Each cell corresponds to an attention head, indexed by its layer (rows) and head position (columns). The Fisher heatmap highlights heads with the largest gradient-based sensitivity, while the $\Delta$CLEVER heatmap indicates robustness changes when individual heads are removed (brighter values correspond to larger robustness gains). Interestingly, Fisher values reveal clear structural patterns: for instance, head 10 consistently exhibits high Fisher importance across layers, suggesting it encodes strongly loss-sensitive features shared throughout the network. By contrast, the $\Delta$CLEVER map highlights different subsets of heads whose removal improves robustness, particularly concentrated in mid-to-late layers.

This contrast underscores the complementary nature of the two metrics: Fisher emphasizes heads that are critical for loss optimization and therefore important for preserving model accuracy, whereas $\Delta$CLEVER identifies heads most relevant for robustness gain. In practice, this complementarity

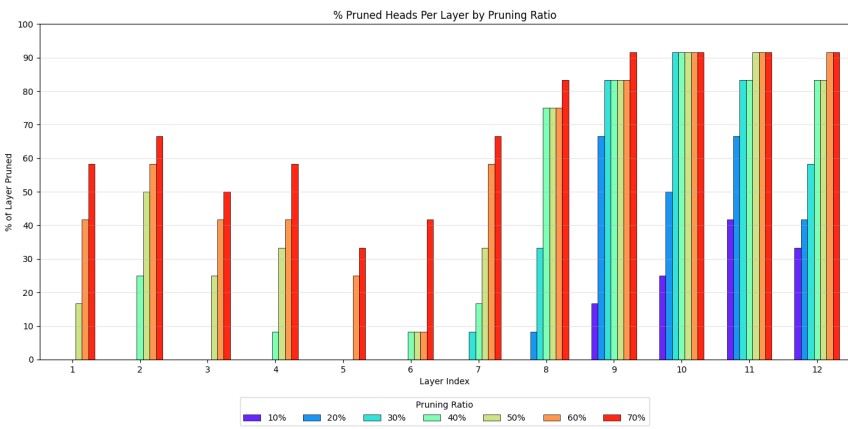

Figure 2: **Distribution of pruned heads per layer in RoBERTa$_{\text{BASE}}$ (SST-2)** across pruning ratios.

ensures that RAHP does not overfit to a single criterion but instead balances accuracy preservation with robustness gains. Thus, we keep heads that appear dark in the Fisher heatmap, indicating high importance for accuracy, and remove heads that appear light in the $\Delta$CLEVER heatmap, as these contribute most to robustness gains.

Figure 4 illustrates the composite pruning scores obtained after combining $\beta = 1$ with $\alpha = 0.5$. The left panel shows the full composite score matrix, while the right panel overlays RAHP's pruning decisions: black crosses mark heads selected for pruning, and red boxes denote heads protected as part of the $10\%$ non-prunable fraction. A clear pattern emerges: most pruning decisions occur in the deeper layers, where many heads exhibit higher composite scores. This trend aligns with prior findings that middle-to-deeper attention heads are more redundant and can be pruned with minimal performance loss (Ling et al., 2024; Zhang et al., 2024; Sajjad et al., 2023; Gromov et al., 2024).

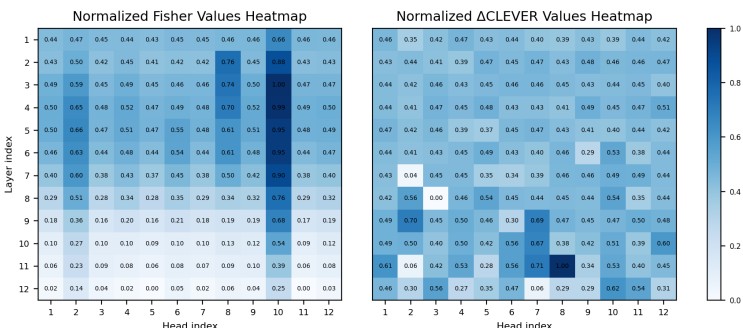

Figure 3: **Norm. Fisher & $\Delta$CLEVER heatmaps for RoBERTa$_{BASE}$ on SST-2.** Fisher shows loss-sensitive heads, while $\Delta$CLEVER highlights robustness-critical mid–late heads, reflecting their complementarity.

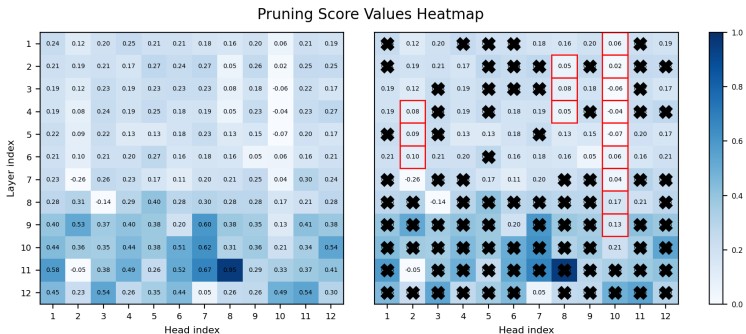

Figure 4: **Composite scores and RAHP decisions for RoBERTa$_{BASE}$ on SST-2.** Left: composite scores ($\beta = 1$, $\alpha = 0.5$). Right: pruning results with crosses for pruned heads and red boxes for the protected set.

## 6 CONCLUSIONS

We introduced RAHP, a *CLEVER-guided, robustness-aware head pruning* framework for Transformer-based text classifiers. RAHP combines a CLEVER-style robustness estimator with Fisher Information to score attention heads, and then performs a single global pruning step that balances robustness gains against accuracy preservation. Across GLUE and AdvGLUE benchmarks, and over multiple backbones, RAHP consistently improves robustness metrics and adversarial accuracy while maintaining competitive clean performance, even at high pruning volumes.

Our analysis shows that RAHP prunes heads in a principled and interpretable way: robustness-critical heads cluster in middle layers, while accuracy-critical heads naturally remain protected through the Fisher term and the non-prunable fraction $\gamma$. Importantly, we use CLEVER as a scalable estimated robustness signal rather than a formal certificate, and our claims focus on improving CLEVER-style robustness metrics and adversarial resistance.

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

# A  ABLATION STUDY

To assess the impact of our scoring function's trade-off parameters, we conduct an ablation study varying the values of $\alpha$ (Fisher-based accuracy cost) and $\beta$ (robustness reward via $\Delta$CLEVER). While our earlier experiments surveyed two other methods, here we focus on two representative models: DeBERTaV3 and DistilBERT, to illustrate how both a high-capacity variant and a lightweight distilled model respond to different $(\alpha, \beta)$ settings. Figure 5 visualizes the results for two models: *DeBERTaV3* and *DistilBERT*. Each point represents a specific $(\alpha, \beta)$ configuration, evaluated by its clean accuracy and CLEVER robustness score.

The results reveal that neither a purely accuracy-driven objective nor a purely robustness-driven objective yields a desirable trade-off. For instance, while $\alpha=0, \beta=1$ achieves the highest CLEVER score, it suffers from a significant drop in accuracy. Conversely, $\alpha=1, \beta=0$ maintains high accuracy but offers minimal robustness gains.

The configuration $\alpha=0.5, \beta=1$ consistently strikes an effective balance across both models, and among all other models presented during the paper, significantly improving robustness over the baseline while preserving accuracy. We adopt this setting for all main experiments, confirming the need to balance accuracy and robustness during pruning.

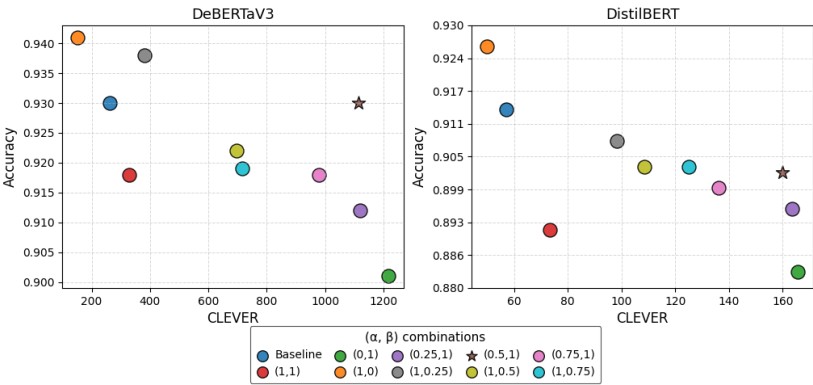

Figure 5: **Effect of** $(\alpha, \beta)$ **weights** on the robustness–accuracy trade-off for DeBERTaV3 (left) and DistilBERT (right). The star denotes the $(0.5, 1)$ configuration used in our main experiments.

