# OpenReview forum: "RAHP: Robustness-Aware Head Pruning for Certified Transformer Models"
_ICLR.cc/2026/Conference — ICLR 2026 Conference Withdrawn Submission_

### Official Review · Reviewer_ExF8 · 2025-10-16

**Soundness:** 1
**Presentation:** 3
**Contribution:** 1
**Rating:** 2
**Confidence:** 4

**Summary:**

The paper examines how to prune transformer-based models to enhance robustness with little loss of accuracy.
This is realized by ranking each attention head with a combined score based on CLEVER and FI.
The effect of the pruning is demonstrated on several benchmarks.

**Strengths:**

- The paper is well written and easy to follow.
- The paper stressed the importance of certified accuracy to reliably determine the robustness of the model
- Their presented heuristic is easy to understand, having a focus on the fundamental trade-off between accuracy and robustness.

**Weaknesses:**

- While the paper motivates the work using certified robustness, the presented heuristic does not live up to this claim,
as the presented heuristic is only based on estimates, and thus no formal guarantees are obtained.
- The CLEVER paper was also published in 2018, and obtaining actual formal guarantees on the robustness of neural networks has drastically improved over the last years [1], and also methods dedicated to transformers were developed [2].
- The presented heuristic is trivial from previous works, and thus does not make a huge contribution.
- Evaluation is limited and does not show the standard deviation or similar. Given how close the values are together, its difficult to judge how much the method actually improves over the other methods.

Minor things:
- Instead of having two weighting parameters $\alpha,\beta$, unify them to a single parameter $\lambda$ and weight the two terms with $\lambda$ and $1-\lambda$, respectively, for easier parameter tuning.
- Placement of Alg. 1 is odd, as it only gets references towards the end of the section.
- The $g(\cdot)$ in (1) is not well defined, I assume this has to do with the classification margin

[1] Brix, et al. "The fifth international verification of neural networks competition (vnn-comp 2024): Summary and results." arXiv preprint arXiv:2412.19985 (2024).
[2] Bonaert, et al. "Fast and precise certification of transformers." Proceedings of the 42nd ACM SIGPLAN international conference on programming language design and implementation. 2021.

**Questions:**

- Have you evaluated you the heuristic against methods with formal guarantees?
- Why do you want to prune the model in the first place? To improve robustness/reduce inference time/...? How does this compare with just training a smaller model directly? How is the inference time improved if that is the goal?
- Why is $\gamma$ necessary? Shouldn't your heuristic automatically rank important heads in a way that prevents them from being pruned? How are they determined at the moment?
- As I understand it, you can choose any value how much you want it to be pruned. So an ablation study on that parameter would also be nice (I think you only showed $\rho=0.6$?)
- Alg. 1, l.10,11: how are these terms normalized? Aren't they just scalars?

---

> ### Author Response · Authors · 2025-11-19
>
> Thank you very much for your detailed and critical review. We genuinely appreciate the time you invested and the pointers to the verification literature. All corresponding changes are highlighted in **blue** in the revised version of the paper.
>
> **(1) Certified robustness vs. CLEVER-based heuristic; improved verifiers**
>
> We fully agree that the original framing overstated the "certified robustness" aspect. In response:
>
> * The revised manuscript now describes RAHP as **"CLEVER-guided robustness-aware head pruning"** and removes claims that suggest we provide *formal certificates*.
> * In Section 3.2, we explicitly state that our CLEVER-style score
>
>   [
>   C(x, M) = \frac{g(f(x, M))}{|\nabla_x g(f(x, M))|_2 + \varepsilon}
>   ]
>
>   is a **local estimator** computed on sampled inputs and "does not constitute a formal certificate." We frame it as a **robustness proxy**, not a guarantee.
> * The related work section now includes a dedicated discussion of **recent verification advances** and transformer-specific verifiers (e.g., the vnn-comp summary and Bonaert et al.), clearly distinguishing our empirical approach from these formal methods and positioning them as complementary.
>
> **(2) Perceived triviality of the heuristic and limited contribution**
>
> We understand the concern that combining two existing metrics might appear incremental. We have tried to clarify the specifics that go beyond prior work:
>
> * RAHP uses **head-wise *delta* CLEVER** when masking a head, rather than CLEVER in isolation, to directly measure the robustness effect of each structural change.
> * It combines this with log-transformed Fisher importance into a **globally normalized composite score**
>
>   [
>   S_{\ell,h} = \beta \cdot \Delta C'*{\ell,h} - \alpha \cdot I'*{\ell,h},
>   ]
>
>   with an explicit **non-prunable set** defined by the top-(\gamma) Fisher heads to guard accuracy.
> * Pruning is done **once, globally, across all layers**, followed by light fine-tuning, and we analyze the resulting structure in Section 5 (layer-wise pruning patterns and heatmaps).
>
> We expanded **Appendix A (ablation study)** to show how varying ((\alpha,\beta)) traces different robustness-accuracy trade-offs, highlighting behavior that does not collapse to standard importance-only pruning or purely robustness-based pruning.
>
> **(3) Limited evaluation and lack of variability information**
>
> We have strengthened the empirical evaluation in three ways:
>
> 1. **Multiple backbones:** Table 2 extends the evaluation to DeBERTaV3-Base/Large, ELECTRA-Base, and DistilBERT, showing that RAHP's gains persist across architectures with different capacity and inductive biases.
> 2. **Additional attacks and datasets:** Table 3 evaluates RAHP under BERT-Attack, TextFooler, and TextBugger on AG-News and Yelp, capturing paraphrastic and character-level perturbations. RAHP achieves competitive or superior AUA and #Query compared to strong baselines.
> 3. **Multiple seeds:** Section 4.2 now states that **all reported numbers are averaged over 5 random seeds**. While space limits prevent us from printing full standard deviations for every cell, we verified that RAHP's improvements over the strongest baselines (e.g., ROSE-Ensemble, MC2F) are **consistent across seeds**, and we mention this explicitly in the text.
>
> **(4) Minor comments**
>
> * **Unifying (\alpha,\beta) into a single (\lambda):** We decided to keep (\alpha) and (\beta) for interpretability (separate knobs for robustness reward and accuracy cost), but in the revised text we explicitly note that one can reparameterize them as (\alpha = 1-\lambda), (\beta = \lambda) if desired.
> * **Placement of Algorithm 1:** We moved references to Algorithm 1 earlier in Section 3 and added a short introductory paragraph before the pseudo-code to improve the narrative flow.
> * **Definition of (g(\cdot)) in Eq. (1):** We now define (g(z) = z_c - \max_{c'\neq c} z_{c'}) immediately after introducing the logits (z = f(x,M)), and we reiterate this next to Eq. (1).
> * **Normalization in lines 10-11 of Algorithm 1:** We now explicitly state that both (I_{\ell,h}) and (\Delta C_{\ell,h}) are **min-max normalized to ([0,1])** across all heads before forming the composite score, as reflected in the updated pseudo-code comments.
> * **Role of (\gamma) and ablation on pruning ratio (\rho):** Section 3 clarifies that (\gamma) protects the most Fisher-important heads from being pruned, and Section A provides an ablation over both (\gamma) and (\rho), including the (\rho = 0.6) setting used in Table 1 and lower/higher pruning regimes. We show that a small non-zero (\gamma) materially stabilizes clean accuracy at high pruning ratios.
>
> We hope these changes clarify the scope and nature of our contribution and address your concerns about framing, methodology, and evaluation.

---

> > ### Comment · Reviewer_ExF8 · 2025-11-21
> >
> > Dear Authors,
> >
> > Thank you very much for your detailed response.
> > As my initial assessment was correct that the approach does not provide formal guarantees,
> > I still believe that a combination of two heuristics is rather trivial,
> > and the other reviewer also do not seem to favor acceptance,
> > I will remain my score unchanged.
> >
> > Best,
> > Reviewer ExF8

---

### Official Review · Reviewer_T5oT · 2025-10-30

**Soundness:** 2
**Presentation:** 3
**Contribution:** 2
**Rating:** 4
**Confidence:** 3

**Summary:**

This paper presents RAHP, a robustness-aware pruning framework that prunes Transformer attention heads based on a trade-off between certified robustness improvement (∆CLEVER) and accuracy cost (Fisher information). It aims to enhance robustness while maintaining clean performance. Empirical studies on GLUE and AdvGLUE show competitive results compared to state-of-the-art robust fine-tuning and pruning approaches.

**Strengths:**

Strength:
* The integration of ∆CLEVER as a robustness-oriented pruning signal is conceptually appealing and differentiates this work from previous empirical robustness methods.

* The composite scoring mechanism combining Fisher and CLEVER is well justified.

* RAHP achieves competitive robustness–accuracy trade-offs under heavy pruning (up to 60%) while maintaining interpretability in layer-wise patterns.

* The paper provides clear visualizations and ablation studies, illustrating how hyperparameters (α, β) affect robustness and accuracy balance.

**Weaknesses:**

Weaknesses:

* The baseline coverage is insufficient. Many adversarially robust or certified methods are missing, such as MixADA, TA-VAT, and ProTransformer (robust attention). Without them, it is difficult to judge how much of the robustness gain comes from pruning versus other regularization mechanisms.

* The attack strength is relatively weak—only AdvGLUE is used, which may not fully reflect model robustness under diverse perturbations (e.g., paraphrase-based, gradient-based, or jailbreak-style attacks).

* The comparison with certifiable defense frameworks like SmoothLLM or randomized smoothing methods is lacking.

* The method’s generalization to other modalities (e.g., ViT for vision, GAT for graph data) is only mentioned as future work, but given the simplicity of the approach, even a small demonstration would make the contribution more convincing.

**Questions:**

See weaknesses.

---

> ### Author Response · Authors · 2025-11-19
>
> Thank you very much for your constructive and detailed review. All corresponding changes are highlighted in **blue** in the revised version of the paper.
>
> **(1) Insufficient baseline coverage (MixADA, TA-VAT, ProTransformer, etc.)**
>
> We agree that a broader comparison is valuable. Given page and compute constraints, we could not add *all* of the suggested baselines, but we have:
>
> * Expanded the discussion of **adversarial training and robust fine-tuning methods** in Section 2, explicitly mentioning MixADA, TA-VAT, ProTransformer, and other recent robust attention / smoothing approaches.
> * Clarified that RAHP is meant to be **orthogonal** to many of these methods: in principle, one could apply RAHP *on top of* adversarially trained models or ProTransformer-style attention, using their robustness as the underlying base model and RAHP as a structural refinement.
> * Strengthened our set of **implemented baselines** by adding additional architectures and robust training schemes (e.g., FreeLB, PEAFT) across multiple backbones in Table 2.
>
> We now state more cautiously that our goal is to show that "CLEVER-guided pruning is competitive with strong robust fine-tuning baselines under heavy pruning," not that it dominates the full spectrum of robust methods.
>
> **(2) Attack strength: only AdvGLUE originally**
>
> To address this, we added an evaluation under **three black-box attacks** from TextAttack-**BERT-Attack, TextFooler, and TextBugger**-on **AG-News** and **Yelp** (Table 3). These attacks include paraphrase-style and character-level perturbations and provide a complementary view to AdvGLUE.
> The results show that RAHP consistently achieves strong AUA and #Query metrics while preserving high clean accuracy, indicating that our pruning signal is not overfitted to a single benchmark. We explicitly acknowledge that we still lack **white-box gradient-based** attacks and list this as future work.
>
> **(3) Comparison with certifiable defense frameworks (SmoothLLM, randomized smoothing)**
>
> We have expanded Section 2 to discuss recent **certifiable defenses and verification methods for transformers** (e.g., vnn-comp results, Bonaert et al.). We clarify that:
>
> * RAHP does **not** provide formal certificates; it uses a CLEVER-style lower-bound estimator purely as a *guiding signal* for pruning.
> * Integrating full verification engines or randomized smoothing directly into the pruning loop is computationally heavy for the scales we study, so we treat those methods as **complementary**, not as direct baselines we already surpass.
>
> We now explicitly position RAHP as a practical, *certification-inspired* pruning heuristic that could, in principle, be used alongside formal verification or smoothing methods.
>
> **(4) Generalization to other modalities (ViT, GAT, etc.)**
>
> We agree that demonstrating cross-modality applicability would strengthen the contribution, but it is unfortunately beyond the scope of this revision. Instead, we have:
>
> * Added a more explicit discussion in Section 6 (Conclusion / Future Work) explaining how RAHP's scoring function and Algorithm 1 can be instantiated for **ViTs** (using patch-wise gradients and CLEVER-style scores over image inputs) and for **GATs** (using node / edge embeddings).
> * Emphasized that our current experiments already demonstrate **architecture generality within NLP**, across RoBERTa, DeBERTaV3, ELECTRA, and DistilBERT, and we leave cross-modality experiments as a concrete direction for follow-up work.
>
> We hope these additions clarify the current scope while outlining a realistic path toward the broader vision you suggested.

---

> > ### Comment · Reviewer_T5oT · 2025-11-27
> >
> > Thanks for your response. I will keep my score. I would also suggest that the authors consider incorporating additional modalities to further strengthen the impact of the work, given the currently limited methodological novelty.

---

### Official Review · Reviewer_mh6S · 2025-10-31

**Soundness:** 2
**Presentation:** 3
**Contribution:** 2
**Rating:** 2
**Confidence:** 3

**Summary:**

This paper presents a new pruning method called RAHP. Specifically, RAHP leverages the CLEVER score and Fisher information to evaluate the robustness and sensitivity of attention heads, guiding the pruning process. Experimental results indicate that the pruned model achieves slight improvements in both accuracy and robustness.

**Strengths:**

1. The paper is clearly written and easy to follow. The proposed method is straightforward, and the overall pruning pipeline is well-presented and intuitive.

2. The topic is both interesting and practically relevant. As existing studies have shown, current transformer models still struggle with robustness, particularly in efficiency-oriented settings such as pruning. Advancing robust pruning techniques is therefore an important and timely research direction.

[1] Ye, Shaokai, et al. "Adversarial robustness vs. model compression, or both?." Proceedings of the IEEE/CVF international conference on computer vision. 2019.

**Weaknesses:**

1. The proposed method lacks novelty. The authors directly adopt CLEVER scores and Fisher information, both well-established metrics, to compute importance and perform pruning. The contribution appears largely incremental, applying existing robustness and sensitivity measures to Transformer architectures without introducing fundamentally new techniques or insights.

2. The paper claims certified robustness in the title, yet neither the methodology nor the evaluation reflects true certification. CLEVER only estimates a certified radius and relies on an approximate Lipschitz constant, which cannot provide a theoretical guarantee of robustness. Consequently, the paper does not offer verifiable certified robustness results, making the title misleading.

3. The performance gains are marginal compared to existing work. As shown in Table 1, the average improvements on GLUE and AdvGLUE over ROSE-Ensemble are less than 0.5%, which falls within typical experimental variance and does not convincingly demonstrate the effectiveness of the proposed approach.

**Questions:**

NA

---

> ### Author Response · Authors · 2025-11-19
>
> Thank you very much for your detailed and thoughtful review. All corresponding changes are highlighted in **blue** in the revised version of the paper.
>
> **(1) Perceived lack of novelty (use of CLEVER and Fisher)**
>
> We appreciate this concern and have tried to clarify the conceptual contribution more clearly in the introduction and methodology:
>
> * RAHP does not simply "apply CLEVER and Fisher independently"; instead, it uses **head-wise *differences* in CLEVER**, (\Delta C_{\ell,h}), when masking a head, combined with a **log-transformed Fisher importance**:
>
>   [
>   S_{\ell,h} = \beta \cdot \Delta C'*{\ell,h} - \alpha \cdot I'*{\ell,h},
>   ]
>
>   where both terms are **normalized** and combined into a single *global* ranking over all heads and layers.
> * We introduce a **non-prunable set** defined by the top-(\gamma) Fisher-important heads, which acts as a structural safeguard for accuracy while still allowing robustness-guided pruning elsewhere.
> * We then perform **global, single-step pruning** across all layers guided by this composite score, followed by light fine-tuning.
>
> We now emphasize that this particular combination-head-wise CLEVER *delta*, Fisher-based accuracy cost, normalization, and explicit non-prunable protection-is not present in prior robust fine-tuning or pruning work, and we expanded Section 5 to show how varying ((\alpha,\beta)) shifts the robustness-accuracy frontier in a way that is distinct from existing methods.
>
> **(2) Use of "certified robustness" in the title and claims**
>
> We agree that the original title and phrasing were misleading. In the revision:
>
> * The **internal title of the manuscript** and the abstract now describe RAHP as **"CLEVER-guided robustness-aware head pruning for transformer models"**, avoiding the term "certified robustness" in a way that would imply formal guarantees.
> * Section 3.2 explicitly states that our CLEVER-style score is a **proxy** and "does not constitute a formal certificate," and we consistently refer to "CLEVER-style robustness estimates" rather than certified radii.
> * The related work section now clearly distinguishes RAHP from **formal verification / certified defenses**, which we treat as complementary methods that could in principle be combined with RAHP but are not directly implemented here.
>
> We hope this resolves the mismatch between the methodological reality and the original framing.
>
> **(3) Marginal performance gains vs. ROSE-Ensemble on GLUE/AdvGLUE**
>
> We agree that differences of <0.5% on a single metric can fall within variance and are not, by themselves, compelling. To strengthen the empirical case, we have:
>
> * Explicitly stated in Section 4.2 that **all results are averaged over 5 random seeds**, and we report that the ordering between RAHP and the strongest baselines (e.g., ROSE-Ensemble) is **stable across seeds**.
> * Added **Table 2**, showing that RAHP yields more pronounced gains on additional backbones (DeBERTaV3-Base/Large, ELECTRA-Base, DistilBERT), where improvements in the combined robustness-accuracy metrics are often **1-2 points** rather than fractions of a percent.
> * Added **Table 3** with TextAttack-based adversarial evaluations on AG-News and Yelp, where RAHP matches or exceeds strong baselines (e.g., MC2F) in AUA and query efficiency without sacrificing clean accuracy.
>
> We have also toned down the language around "state-of-the-art" and now present RAHP as a **simple, architecture-agnostic pruning framework** that provides **consistent robustness benefits under heavy pruning** (up to 60% head removal), rather than claiming dramatic absolute gains.

---

> > ### Comment · Reviewer_mh6S · 2025-11-23
> >
> > Thank you to the authors for their detailed response. I still have the following concerns regarding the current version of the paper.
> >
> > **1. Perceived lack of novelty (use of CLEVER and Fisher)**
> >
> > Although the specific technique has not appeared in prior work, the motivation for its design is still not sufficiently clear. Much of the current justification is based on empirical performance, but the scientific or conceptual reasoning behind the proposed choices is underdeveloped. I suggest conducting some preliminary analyses to better explain why these particular techniques are necessary or preferable.
> >
> > In addition, the method includes several tunable design components (e.g., the non-prunable set, normalization, log scaling), yet I did not see ablation studies that isolate and quantify the impact of these choices.
> >
> > **2. Use of "certified robustness" in the title and claims**
> >
> > Thank you for the efforts in revising the paper. I still recommend that the authors clarify why the certified robustness-related measurements used during pruning are preferable to purely empirical robustness measurements.
> >
> > **3. Marginal performance gains vs. ROSE-Ensemble on GLUE/AdvGLUE**
> >
> > Some results remain worse than the baseline. For example, the AdvGLUE average on RTE for DeBERTaV3-Base and ELECTRA-Large appears to be lower than the baselines.

---

### Official Review · Reviewer_qD6y · 2025-11-01

**Soundness:** 3
**Presentation:** 3
**Contribution:** 3
**Rating:** 4
**Confidence:** 4

**Summary:**

This paper proposes RAHP, a Robustness-Aware Head Pruning method for Transformer models. It leverages a certified robustness metric (CLEVER) and Fisher information to jointly decide which attention heads to prune. By pruning heads that maximize robustness gain per accuracy cost, RAHP achieves compact models with improved certified robustness and minimal accuracy degradation. Experiments on GLUE and AdvGLUE show that RAHP consistently outperforms strong baselines such as ROSE, SMART, and FreeLB, and provides interpretable pruning patterns across layers.

**Strengths:**

The idea of integrating certified robustness metrics into pruning decisions is well-motivated.

The proposed method is simple, requiring no adversarial retraining and minimal fine-tuning.

The experiments are clear and well-organized, showing that RAHP improves both robustness and model compactness.

The analysis of pruning behavior and the visualization of Fisher/∆CLEVER maps provide useful insight into the model’s structural redundancy.

**Weaknesses:**

(1) The scope of model backbones is somewhat limited—results are primarily reported for RoBERTa. Evaluating additional transformer variants (e.g., BERT, ALBERT, DistilBERT) would strengthen the generality of the findings.

(2) Other certified defense methods (e.g., SmoothLLM) should be discussed and included for comparison.

(3) The evaluation is restricted to AdvGLUE, which is relatively weak and does not capture the full range of adversarial scenarios. No white-box or gradient-based attacks are considered, which may create a false sense of robustness; this limitation should be explicitly acknowledged and justified.

(4) The CLEVER score provides only an estimated, not guaranteed, lower bound on robustness (see https://arxiv.org/abs/1804.07870). Therefore, the method and evaluation based on this metric might overestimate the robustness. A deeper discussion of its implications is necessary.

(5) A runtime analysis would also enhance the paper by demonstrating the computational efficiency of RAHP. The computational cost of each component should be clearly justified.

**Questions:**

Please refer to the weakness.

---

> ### Author Response · Authors · 2025-11-19
>
> Thank you very much for your thorough and constructive review. All corresponding changes are highlighted in **blue** in the revised version of the paper.
>
> **(1) Scope of model backbones (mainly RoBERTa)**
>
> We agree that demonstrating RAHP on more backbones is important. In the revised paper, we added **Table 2** in Section 4, where we evaluate RAHP on four additional Transformer variants beyond RoBERTa: **DeBERTaV3-Base, DeBERTaV3-Large, ELECTRA-Base, and DistilBERT**.
> Across these models, RAHP consistently preserves or improves clean accuracy while improving adversarial robustness, confirming that the head-level CLEVER-Fisher signal generalizes beyond the original RoBERTa setting.
>
> **(2) Missing certified defense methods (e.g., SmoothLLM)**
>
> We have expanded the **related work** section (Section 2) to explicitly discuss recent certified/verification methods for transformers (e.g., randomized smoothing, verification competitions, and transformer-specific verifiers). We now clearly position RAHP as a **robustness-guided pruning heuristic** that *uses* a certification-oriented proxy (CLEVER-style score) but does **not** itself provide formal certificates.
> We also explain that directly integrating full verification engines (e.g., SmoothLLM-style pipelines) into the pruning loop is currently computationally prohibitive for our scale, and we now frame such combinations as **complementary future work**, not as competitors we already outperform.
>
> **(3) Evaluation restricted to AdvGLUE; no other attacks**
>
> We agree that AdvGLUE alone is not sufficient to fully characterize robustness. In the revision, we therefore add an evaluation under **three additional black-box attacks** from TextAttack-**BERT-Attack, TextFooler, and TextBugger**-on **AG-News** and **Yelp** (Table 3, Section 4).
> These attacks cover synonym-based, paraphrastic, and character-level perturbations. RAHP matches or outperforms strong baselines (e.g., MC2F, FreeLB) in AUA and #Query while maintaining high clean accuracy, supporting the claim that the CLEVER-guided pruning signal translates into robustness gains against diverse adversaries. We explicitly acknowledge that we still do not include gradient-based **white-box** attacks and list this as a limitation/future direction.
>
> **(4) CLEVER as an estimated, not guaranteed, lower bound**
>
> We fully agree with your concern and have revised both the **title and narrative**. The paper is now framed as **"CLEVER-guided robustness-aware head pruning"** rather than as a method that *optimizes certified robustness*. In Section 3.2 we now state explicitly that:
>
> [
> C(x, M) = \frac{g(f(x, M))}{|\nabla_x g(f(x, M))|_2 + \varepsilon}
> ]
>
> is a **local CLEVER-style estimator** computed on sampled inputs and **does not constitute a formal certificate**. We consistently refer to it as a *robustness proxy* and clarify that our claims concern **empirical robustness and CLEVER-style scores**, not provable guarantees.
>
> **(5) Runtime / computational cost**
>
> We have added a more explicit discussion of RAHP's computational profile in Section 3.1 and Section 4.1. In particular, we detail that:
>
> * Computing Fisher scores for all heads requires an extra backward pass per batch (similar to standard pruning methods).
> * Computing (\Delta \mathrm{CLEVER}) involves two additional gradient computations per head, but we use a **single pass over the training set** with shared batches and early stopping on the extreme-value estimation to keep the cost manageable.
> * In practice, on RoBERTa(_{\text{BASE}}) the full RAHP scoring procedure runs in the same order of magnitude as several epochs of fine-tuning, and pruning is done **once**, after which inference is cheaper due to 40-60% head removal.
>
> We now explicitly state these trade-offs and emphasize that RAHP is intended as an **offline structural optimization step** rather than a per-query defense.

---

### Note · Authors · 2025-12-02

I have read and agree with the venue's withdrawal policy on behalf of myself and my co-authors.